# A Study on the Interpretability of Diabetic Retinopathy Diagnostic Models

**DOI:** 10.3390/bioengineering12111231

**Published:** 2025-11-10

**Authors:** Zerui Zhang, Hongbo Zhao, Li Dong, Lin Luo, Hao Wang

**Affiliations:** 1School of Bioengineering, Chongqing University, Chongqing 400044, China; 30026394@alu.cqu.edu.cn (Z.Z.); 202319131108@stu.cqu.edu.cn (H.Z.); 2Beijing Tongren Eye Center, Beijing Tongren Hospital, Capital Medical University, Beijing 100730, China; dongli@ccmu.edu.cn; 3College of Engineering, Peking University, Beijing 100871, China; 4Institute for Medical Device Control, National Institutes for Food and Drug Control, Beijing 102629, China

**Keywords:** diabetic retinopathy classification models, model interpretability, fundus images, interpretability evaluation

## Abstract

This study focuses on the interpretability of diabetic retinopathy classification models. Seven widely used interpretability methods—Gradient, SmoothGrad, Integrated Gradients, SHAP, DeepLIFT, Grad-CAM++, and ScoreCAM—are applied to assess the interpretability of four representative deep learning architectures, VGG, ResNet, DenseNet, and EfficientNet, on fundus images. Through saliency map visualization, perturbation curve analysis, and trend correlation analysis, combined with four quantitative metrics—saliency map entropy, AOPC score, Recall, and Dice coefficient—the interpretability performance of the models is comprehensively assessed from both qualitative and quantitative perspectives. The results show that model architecture greatly influences interpretability quality: models with simpler structures and clearer feature extraction paths (such as VGG) perform better in terms of interpretability, while deeper or lightweight architectures exhibit certain limitations.

## 1. Introduction

Diabetic retinopathy (DR), the most prevalent microvascular complication of diabetes, has become one of the leading causes of blindness among adults worldwide [1]. Epidemiological data indicates that approximately 20% of the global 463 million diabetic patients are at risk of developing DR [2]. Characterized by progressive progression, DR can lead to irreversible vision impairment or complete blindness if not diagnosed and treated promptly. Therefore, early screening and diagnosis hold critical clinical significance for disease prevention and control. However, current clinical DR screening primarily relies on manual evaluation of fundus images by specialized ophthalmologists—a process that is both time-consuming and subjective. Particularly in underserved areas with limited ophthalmic resources, this approach struggles to meet growing screening demands. Moreover, traditional manual assessment methods exhibit notable limitations in accuracy, consistency, and scalability. To address these challenges, the academic community has recently developed deep learning-based automated DR classification systems like [3,4,5,6] by leveraging large-scale public datasets of fundus color images. Through end-to-end trained neural network models, these systems enable reliable identification and grading of retinal lesions, providing robust technical support for large-scale screening initiatives.

Model interpretability is not commonly listed as a research priority; researchers often focus more on algorithm optimization or cross-domain applications. Such technological breakthroughs can effectively improve model performance metrics, aligning with scientists’ goals of pursuing technological innovation. Model interpretability research has faced significant challenges and delays. The lack of intuitive metrics like accuracy rates hinders the immediate demonstration of research value. Moreover, deep learning models—particularly complex multi-layer neural networks—present substantial difficulties in analyzing their decision-making mechanisms. This oversight in medical practice reveals critical flaws: clinicians struggle to understand diagnostic rationale through the “black box” nature of these models, and are often unable to identify key image features underlying lesion classification results. This undermines both the credibility and practical application of models in healthcare. Even when models demonstrate high theoretical diagnostic accuracy, the absence of interpretability makes them less trustworthy for doctors, ultimately limiting their potential as effective diagnostic aids.

The purpose of conducting interpretable analysis on models is to break down their “black box” characteristics and reveal the underlying logic in processing retinal image data and generating lesion grading predictions. For diabetic retinopathy (DR) classification models, good interpretability means the model can clearly summarize lesion features for DR and establish a close connection with medically defined DR diagnostic principles. This interpretability not only helps researchers deeply understand the internal mechanisms and working principles of the model, thereby more accurately evaluating its effectiveness, reliability, and potential risks in DR grading, but also provides strong support for developers to optimize model structures and improve algorithm design. In the medical field, especially for DR diagnosis where it directly affects patient health and treatment plan formulation, the requirements for model safety and transparency are extremely high, making interpretability particularly crucial. Therefore, making the “black box” of deep learning in DR classification models transparent and assessing model interpretability holds significant importance.

The main contributions of this work are summarized as follows:

To effectively evaluate the interpretability of DR grading models, this study constructed a clinically annotated dataset comprising 2060 fundus images collected from six local hospitals. Each image was independently annotated by at least two professionally trained ophthalmologists, with labeling encompassing both disease severity grading and pixel-level delineation of lesion areas—including hard exudates, hemorrhages, microaneurysms, and soft exudates—using polygonal bounding boxes.

This study applied multiple mainstream interpretability analysis methods to generate saliency maps illustrating model decision processes. Initial qualitative validation was performed using perturbation curve analysis to assess whether the highlighted regions in these maps accurately reflect areas critical to model predictions. Furthermore, the clinical alignment of model reasoning was rigorously evaluated through trend correlation analysis, which verifies whether the model’s focus shifts consistently with DR severity progression in a manner consistent with clinical grading logic. The quality of saliency maps was quantitatively measured using entropy-based evaluation, where lower entropy indicates more focused and interpretable attention regions, while higher entropy suggests dispersed or noisy explanatory signals.

Experimental results demonstrate that model architecture significantly influences interpretability quality. Models with simpler structures and clearer feature extraction paths, such as VGG, exhibit superior interpretability performance, while deeper or lightweight architectures show certain limitations in explanation clarity and consistency. This research explored effective strategies for enhancing interpretability in DR classification models across different architectures, providing both theoretical insights and practical references for developing clinically applicable AI-assisted diagnostic systems.

The remainder of this paper is structured as follows. Section 2 reviews related work. Section 3 introduces the experimental materials and methods. Section 4 presents the experimental results. Section 5 provides a discussion. Section 6 concludes the paper.

## 2. Related Research

To ensure the comprehensiveness and inclusiveness of the review, literature retrieval was primarily conducted in common academic databases including Web of Science Core Collection, IEEE Xplore, PubMed, and Scopus. For the retrieval strategy, we adopted highly relevant keyword combinations related to the interpretability of diabetic retinopathy classification models, mainly including: “diabetic retinopathy classification,” “deep learning,” “interpretability,” “explainable AI,” “saliency map,” “Grad-CAM,” “SHAP,” “feature visualization,” etc. The search queries were optimized using Boolean operators (such as AND, OR) to balance the precision and coverage of retrieval results.

To strictly screen relevant literature, we established clear inclusion and exclusion criteria. The included literature had to meet the following conditions: (1) the research content directly involves deep learning-based DR grading or classification models; (2) the study explicitly discusses or practically applies model interpretability methods; (3) the literature is published in peer-reviewed academic journals, conference proceedings, or recognized preprint platforms (such as arXiv). Simultaneously, we excluded the following types of literature: (1) literature not written in English; (2) studies focusing only on binary classification DR detection rather than multi-class grading; (3) literature with incomplete method descriptions or unavailable full text. Through the above criteria, we ensured the relevance, academic standardization, and content completeness of the reviewed literature.

In literature introducing DR classification models, model interpretability has rarely been prioritized as a research focus. Researchers tend to concentrate on algorithmic improvements or cross-domain applications. Algorithmic enhancements can effectively boost model performance metrics, aligning with scientists’ pursuit of technological breakthroughs. For instance, Firke S N et al. introduced an improved convolutional neural network method for DR detection, utilizing RMSprop optimizer during training to achieve 96.15% classification accuracy on the APTOS dataset [7]. Cross-domain applications rapidly expand model applicability by exploring generalization capabilities across diverse retinal image datasets. Kobat S G et al., for example, applied pre-trained DenseNet201 on the ImageNet1k dataset to APTOS 2019 Blindness Detection dataset, achieving 87.43% accuracy in five-category classification [8]. Asia A O et al. fine-tuned ResNet-101, ResNet-V1-50, and VGG-16 models on DR datasets, ultimately attaining significant performance improvements with 98.82% and 91.5% accuracy on ResNet-101 and ResNet-V1-50, respectively [9]. Furthermore, the research by Kao Y. H et al., which employs the EfficientNet architecture for transfer learning, achieved 84% accuracy in multi-category classification and 99% accuracy in binary classification on a large-scale annotated retinal image dataset. Their performance outperformed multiple common CNN models, including VGG16-fc1, VGG16-fc2, NASNet, Xception, Inception ResNetV2, InceptionV3, MobileNet, and ResNet50 [10].

To effectively evaluate the interpretability of DR grading pre-trained models, it is crucial to thoroughly understand current research status and commonly used metrics. Interpretability remains a persistent challenge in deep neural network research, as its fundamental principles have yet to form a widely accepted theoretical framework. While interpretability measurement methods continue evolving with progress, there are multiple feasible approaches to achieve interpretability in deep learning for DR grading models. First, feature visualization [11]: By presenting features learned by the model from retinal images through intuitive visualizations, medical researchers can better comprehend how retinal data is processed and transformed within the model, thereby understanding how it extracts lesion characteristics. Second, rule-based interpretation [12]: Attempting to extract comprehensible rules similar to those followed by medical experts during diagnosis, providing logical justification for model decisions. Third, sensitivity analysis [13]: Determining the significance and impact of input retinal image features on lesion grading outputs, assisting clinicians in assessing the model’s sensitivity to different lesion characteristics.

Explanatory analysis methods can be categorized into black-box and white-box approaches. Black-box analysis primarily targets models with inaccessible internal structures or parameters, interpreting decision-making processes through examining input-output relationships. Common black-box methods include Gradient Sensitivity [14], SmoothGrad [15], Integrated Gradients [16], SHAP [17], DeepLIFT [18], and LIME [19]. The Gradient method quantifies model sensitivity by calculating gradients between outputs and inputs, identifying key features that influence final predictions. Its simplicity and intuitive interpretation make it widely used for visualizing models. However, despite its effectiveness in many tasks, gradient noise often requires optimization. In gradient visualization, SmoothGrad and Integrated Gradients offer enhanced analysis. By applying random noise to input samples and averaging multiple gradient calculations, SmoothGrad reduces high-frequency noise interference, producing smoother and more stable visualizations that significantly improve feature attribution accuracy. Integrated Gradients employs an integral gradient approach, measuring the contribution of input features to final decisions by accumulating gradients from baseline inputs to actual inputs. While this method provides a global interpretation perspective, its computational overhead remains relatively high due to the need for multiple integration points. Compared with Gradient, SmoothGrad, and Integrated Gradients approaches, SHAP and DeepLIFT focus on feature attribution analysis. SHAP, grounded in Shapley value theory, achieves precise and consistent feature importance allocation by calculating marginal contributions for each feature. This method ensures fair computation of each feature’s contribution across all possible combinations, though its computational complexity increases significantly due to requiring traversal of all feature combinations. DeepLIFT adopts a contribution decomposition approach, evaluating feature influence by comparing actual activation values of input features with variations in baseline inputs. While demonstrating higher computational efficiency and effectively capturing nonlinear feature impacts, its result reliability partially depends on the selection of baseline inputs. LIME, a model-agnostic local interpretable method, operates by generating perturbation samples within the local neighborhood of the sample under interpretation. It records the original model’s predictions for these perturbations and then fits a simple, interpretable model (e.g., linear regression) in that region to approximate the local behavior of the original model. This approach can help users understand the decision logic of the model near specific samples. These methods reveal the decision mechanism of the model from different perspectives, and show their own advantages under various tasks and model architectures.

In contrast, white-box analysis methods are particularly effective for models with direct access to internal structures or parameters. By examining the underlying mechanisms of these models, they provide detailed and transparent explanations of decision-making processes, making them ideal for scenarios requiring deep understanding of model operations. Common white-box analysis techniques include Activation Maximization [20], Class Activation Mapping (CAM) [21], Layer-wise Relevance Propagation (LRP) [22], and Decision Tree Interpretation (DTI) [23]. The Activation Maximization method optimizes input images to maximize neural activation in specific neurons, visualizing the feature patterns these neurons respond to. This approach helps researchers understand feature representations learned at particular layers, offering intuitive insights into the model’s internal workings. CAM and its variants (e.g., Grad-CAM [24]) generate heat maps by weighting convolutional layer feature maps, highlighting regions that contribute most significantly to classification decisions. Particularly effective in computer vision tasks, this method visually demonstrates which image regions the model prioritizes during processing. Layered Relevance Propagation (LRP) generates interpretive diagrams by decomposing model decisions into input feature contributions through a hierarchical backpropagation model output. This method provides a complete interpretation pathway from outputs to inputs, making it suitable for scenarios requiring step-by-step analysis of decision-making processes. For decision tree-based models, the decision tree interpretation approach analyzes nodes and branches to explain decision-making logic, particularly effective for ensemble learning methods like random forests and gradient boosting trees. These white-box analysis approaches each have unique strengths and limitations, catering to different model architectures and task scenarios. In practice, researchers typically select appropriate methods based on specific needs or combine multiple approaches for comprehensive insights. In medical imaging analysis, combining CAM with LRP can simultaneously provide explanations for local feature attention and global decision pathways. By integrating black-box and white-box analysis methods, we can more effectively reveal deep learning models ‘internal mechanisms, enhancing their credibility and practical value in critical fields like medical diagnosis. This multi-layered, multi-perspective interpretation strategy not only strengthens clinicians’ trust in model decisions but also offers crucial guidance for model optimization and improvement.

## 3. Materials and Method

To conduct interpretability analysis of diabetic retinopathy (DR) classification models, this study established a comprehensive experimental framework encompassing dataset construction, model selection, interpretability analysis, and evaluation. For the dataset, a total of 2060 fundus images were collected from six hospitals. All images were deidentified prior to data collection. Each image was independently annotated by at least two professionally trained ophthalmologists. In addition to disease grading, polygonal bounding boxes were used to outline lesion areas, including hard exudates, hemorrhages, microaneurysms, and soft exudates, providing pixel-level annotations for model interpretability evaluation. In terms of model selection, four representative architectures and their variants—VGG, DenseNet, ResNet, and EfficientNet—were chosen as the classification models for analysis. These models cover a spectrum from shallow networks with simple linear stacking, to deep networks with dense or cross-layer connections, and further to highly optimized lightweight networks. This selection comprehensively reflects the potential impact of different model depths, connection mechanisms, and feature fusion strategies on interpretability. For interpretability analysis, multiple interpretability methods were applied, including gradient-based techniques (Gradient, SmoothGrad, Integrated Gradients), Shapley value-based methods (SHAP), backpropagation-based attribution methods (DeepLIFT), and activation mapping methods (Grad-CAM++, ScoreCAM). The combination of these analytical approaches enables a multi-faceted analysis of model decisions from complementary dimensions such as input sensitivity, feature contribution distribution, attribution propagation mechanisms, and visual region saliency. Finally, for the qualitative and quantitative assessment of the generated saliency maps, this study employed perturbation-based curve analysis and trend correlation analysis to evaluate whether the highlighted regions contribute meaningfully to model decisions and to examine the consistency between model reasoning and clinical logic. Additionally, the entropy, recall, and Dice coefficient of the saliency maps were calculated to measure the concentration of explanatory signals and their spatial overlap accuracy with ground-truth lesion regions.

### 3.1. Dataset Preparation

To analyze the interpretability of the diabetic retinopathy (DR) classification model, this study employed a retrospective random sampling method to collect a total of 2060 fundus images from ophthalmology departments at six tertiary hospitals across five different provinces in China. Geographically, the sample included three hospitals in Beijing, and one hospital each in Northeast, Southeast, and South China. It should be specifically noted that this study utilized fully anonymized retrospective data, and the original data did not contain patients’ age, gender, or other personal information. After excluding 343 images with other ocular diseases, 1717 fundus images were ultimately selected as the experimental dataset for evaluating the interpretability of classification models. To ensure annotation accuracy and reliability, each image was jointly annotated by at least two professionally trained ophthalmologists during data collection. As shown in Figure 1, the annotation included two components: disease grading and lesion area demarcation. Disease severity was categorized into five levels: normal, mild non-proliferative diabetic retinopathy (mild NPDR), moderate NPDR, severe NPDR, and proliferative diabetic retinopathy (PDR). Additionally, each image was marked with polygonal annotation boxes for lesion areas, specifically including key lesion types such as hard exudates, hemorrhages, microaneurysms, and soft exudates.

As shown in Table 1, the distribution of diseased samples within the dataset is relatively balanced. The “Normal” category accounts for 51.02%, followed by “Mild NPDR” (9.73%), “Moderate NPDR” (17.13%), “Severe NPDR” (10.6%), and “PDR” (11.47%). While normal images constitute half of the dataset, the remaining categories maintain a roughly even distribution. Although the sample sizes for “Mild NPDR” and “Severe NPDR” are relatively small, these categories still provide sufficient data to meet testing requirements for evaluating model performance across different DR analysis levels. To facilitate data loading and processing, all fundus images are stored in JPG or JPEG formats.

### 3.2. Preparation of the Models to Be Tested

In order to analyze the interpretability of DR classification model, this study selects four deep learning network architectures—VGG [25], DenseNet [26], ResNet [27] and EfficientNet [28], and trains a total of 16 DR classification models, which are then tested on experimental dataset.

In recent years, network architectures such as VGG, DenseNet, ResNet, and EfficientNet have demonstrated remarkable performance in image recognition tasks, each featuring unique structural designs and advantages that represent different developmental stages of neural networks. The VGG network, renowned for its simplicity and high modularity, primarily extracts image features through stacking multiple 3 × 3 convolutional kernels and 2 × 2 max pooling layers. Although it requires substantial parameters and computational complexity, its unified convolutional architecture facilitates transferability and scalability, making it a widely adopted foundational template for subsequent network architectures across various visual tasks. DenseNet introduces dense connection mechanisms, enabling feature-level connections between all preceding layers, which significantly enhances feature reuse efficiency and gradient flow performance. This connection approach not only effectively mitigates the common gradient vanishing problem in deep networks but also drastically reduces parameter redundancy, allowing models to maintain strong expressive capabilities with fewer parameters. ResNet addresses the performance degradation issue during deep network training by introducing residual connections. Its basic unit employs identity mappings to directly skip-connect inputs to outputs, ensuring stable information flow within the network. This design enables more layers to be stacked without causing performance degradation, leading to groundbreaking advancements in multiple image recognition tasks. EfficientNet is a lightweight convolutional neural network architecture optimized through composite scaling strategies. By holistically considering network depth, width, and image resolution, it automatically determines the optimal scaling ratio to maximize performance under computational constraints. Combining low model complexity with excellent classification accuracy, EfficientNet proves ideal for efficient deployment scenarios. These models span from shallow networks with simple linear stacking, to deep networks employing dense or cross-layer connections, and further extend to highly optimized lightweight networks. This selection comprehensively captures the potential impacts of varying model depths, connection mechanisms, and feature fusion strategies on interpretability.

In order to evaluate the interpretability of different networks and structures in DR classification tasks, as shown in Table 2, this study selected a number of representative network variants as test models based on four mainstream deep learning network architectures.

Variants within different architectures differ in network depth, structural complexity, and regularization strategies. The primary distinction between VGG16 and VGG19 lies in the number of stacked convolutional layers. VGG16 contains 13 convolutional layers and 3 fully connected layers, while VGG19 introduces additional convolutional layers within each convolution block, resulting in a deeper overall structure that theoretically enhances feature extraction capabilities. Additionally, VGG16-BN and VGG19-BN incorporate Batch Normalization (BN) layers into their original architectures, which helps accelerate model convergence and improve training stability. Variants of the DenseNet architecture primarily adjust the number of network layers and the depth of dense blocks to create models like DenseNet121, 161, 169, and 201. These variants maintain the dense connection mechanism while progressively increasing the number of convolutional layers to enhance network expressiveness. Larger numbers indicate deeper networks, with corresponding increases in model parameters and computational costs, though they also improve the modeling capability for fine-grained features. In the ResNet series, ResNet18 and ResNet34 are shallow variants constructed using basic residual units, suitable for small to medium-scale data tasks. In contrast, ResNet50 and ResNet152 belong to deep networks that incorporate bottleneck blocks to control parameter scale while enhancing nonlinear modeling capabilities. As network depth increases, model expressiveness improves significantly, but training strategies and data requirements also rise accordingly. The EfficientNet series employs composite scaling strategies, systematically adjusting network depth, width, and input image resolution to achieve efficient performance scalability. Variants B0 through B3 represent progressively expanded network architectures. With increasing numbering, these variants demonstrate enhanced classification accuracy and improved perception of complex patterns.

All models in Table 2 were trained using the PyTorch 2.1.1 deep learning framework in this experiment. As a widely adopted deep learning platform, PyTorch provides robust tools and modules that enable flexible network design and efficient training processes. To ensure models effectively capture data features and achieve good generalization capabilities, all models were initialized with parameters pre-trained on the ImageNet large-scale image dataset [29]. Each model was fine-tuned on the dataset constructed in this study, which was randomly split into 80% for training and 20% for testing. After 50 training epochs on the training set, model performance was evaluated on the validation set every two epochs. To ensure the comparability and fairness of the experimental results, all models adopted a unified set of hyperparameters in the cross-comparative analysis. Specifically, the configurations included an initial learning rate of 0.0001, a batch size of 32, and the Adam optimization algorithm.

### 3.3. Method

The clinical grading of diabetic retinopathy (DR) is typically determined based on the type of lesions and their distribution. The International Clinical Diabetic Retinopathy (ICDR) severity scale [30], one of the most widely adopted standards, categorizes DR into five grades: Grade 1 (no apparent retinopathy); Grade 2 (mild non-proliferative DR, characterized only by microaneurysms); Grade 3 (moderate non-proliferative DR, with more lesions than microaneurysms alone but insufficient to meet the criteria for severe non-proliferative DR); Grade 4 (severe non-proliferative DR, characterized by: (i) ≥20 intraretinal hemorrhages in each of the four quadrants; (ii) venous beading in two or more quadrants; (iii) prominent intraretinal microvascular abnormalities [IRMA] in at least one quadrant without neovascularization); and Grade 5 (proliferative DR, defined by the presence of neovascularization or vitreous/preretinal hemorrhage [31]). These criteria indicate that the essence of DR grading lies in identifying typical lesions as well as evaluating their quantity and spatial distribution patterns.

The saliency map derived from interpretable analysis methods reveals how different models measure and calculate the contribution of input features. Models with strong interpretability are expected to demonstrate that their interpretation methods highlight regions consistent with critical lesion areas, reflecting alignment between the model’s decision-making process and clinical logic. Furthermore, if changes in lesions observed in fundus images correspond to trends in highlighted regions of the saliency map, it would demonstrate the model’s interpretability and clinical credibility. Evaluating the interpretability of DR classification models involves assessing whether their decision-making rationale aligns with clinical grading logic, as well as evaluating the quality of saliency maps generated through interpretability methods.

To achieve the aforementioned objectives, this paper designs an experimental protocol. As shown in Figure 2, the first phase constitutes the preparation stage, which includes the test model, test images, image masks, and corresponding label information. All test models are stored in.pth format and loaded via the PyTorch framework for inference operations. Test images and their corresponding mask images are read as.jpg files and uniformly resized to 224 × 224 resolution to meet model input requirements. The labels (denoted as DR) consist of five graded categories: normal, mild NPDR, moderate NPDR, severe NPDR, and PDR. To accommodate model evaluation formats, these labels are converted into One-Hot encoding with values ranging from 0 to 4. Test images are selected from the dataset constructed in this study, with five representative images chosen for each of the five lesion grades to ensure diagnostic feature representation and minimal noise interference. Figure 3 displays the selected image samples arranged from top to bottom: normal, mild NPDR, moderate NPDR, severe NPDR, and PDR. Each pathological grade contains five fundus images.

In the second step, we utilized seven mainstream interpretable analysis methods (Gradient, SmoothGrad, Integrated Gradients, SHAP, DeepLIFT, Grad-CAM++, and ScoreCAM) to generate saliency maps for each model, enabling visualization of model focus areas. To ensure accuracy and stability of interpretation results, this study did not adopt the LIME method. While LIME typically relies on superpixel segmentation and region masking for local perturbations in image tasks, followed by linear model fitting to approximate the original model’s local behavior, its segmentation strategy struggles with small, poorly defined lesion areas in diabetic retinopathy images. This makes it difficult to precisely locate critical pathological features, potentially leading to information loss or incorrect segmentation. Furthermore, the method’s reliance on high randomness in perturbation sample generation results in poor stability and reproducibility of interpretation outcomes, failing to meet medical requirements for consistency and reliability in explainable results.

Figure 2’s Stage Two summarizes the applicable conditions for each method. All methods require specifying labels for test images to guide models in generating decision features corresponding to categories. Notably, Integrated Gradients, SHAP, and DeepLIFT also require a baseline input. Differences in theoretical foundations and computational approaches result in variations in implementation. As shown in Table 3, Gradient calculates gradients between input features and model outputs to reveal feature influence on decision-making. This method has low computational complexity, requires no image category designation, and doesn’t depend on baseline inputs. SmoothGrad applies random noise to input samples and calculates gradient averages to reduce high-frequency noise in gradient computation, thereby enhancing feature attribution stability. While not requiring baseline inputs, this method demands image category information and higher computational costs. Integrated Gradients employs integral gradient calculation, measuring cumulative feature contributions through gradient accumulation from baseline to actual inputs. This approach necessitates selecting appropriate baseline inputs and specifying image categories, leading to high computational requirements. Both SHAP and DeepLIFT compute feature importance based on baseline inputs: SHAP uses Shapley values to allocate feature contributions, while DeepLIFT performs attribution analysis by referencing activation values. Both methods require specifying image category information and reference inputs, making them suitable for various tasks with high and medium computational complexity, respectively. Grad-CAM++ and ScoreCAM are both visualization approaches based on convolutional feature maps, primarily applicable to models with convolutional architectures. Grad-CAM++ employs gradient weighting to calculate the influence of feature maps on prediction results, which is ideal for handling complex target objects or multi-instance scenarios. ScoreCAM enhances stability by treating feature maps as masks that repeatedly interact with original images, observing model predictions to assess feature map importance without requiring gradient calculations.

The third phase involves preliminary qualitative evaluation of the saliency map using perturbation curve analysis to assess whether it accurately reflects the model’s critical regions. This method simulates gradual loss of image area information, observing how model predictions evolve to determine the relative importance of saliency map-identified regions in decision-making. The perturbation curve generation process is as follows: For a test image, first rank all pixels by saliency values from highest to lowest to identify the model’s most focused areas. Then, at each perturbation ratio (0–100%, with 2% increments), sequentially mask high-salience regions ranked higher in the saliency list. At each ratio, record the model’s confidence in predicting original categories after masking corresponding high-salience areas. Finally, plot the model’s output probabilities against perturbation ratios on the perturbation curve. If the saliency map accurately identifies the model’s primary focus areas, increased masking should lead to a sharp decline in target category confidence, showing a monotonically decreasing trend with rapid rate of decrease. Conversely, if the masked high-salience regions have minimal impact on predictions, indicating poor representation of actual key areas, the perturbation curve will show no significant change.

The fourth step involves calculating the entropy value of the saliency map and conducting trend correlation analysis to verify whether the model’s decision-making basis aligns with clinical classification logic. The saliency map entropy value measures the concentration of regions of interest within the model, while entropy reflects the randomness or complexity of the saliency map. Higher entropy values indicate scattered distribution of highlighted areas, suggesting unfocused model attention and poor interpretability. Conversely, lower entropy values demonstrate concentrated saliency regions with clear boundaries, indicating more focused model attention and enhanced interpretability and consistency. For saliency maps of images within the same category, similar pattern recognition and regional distribution should be observed—meaning higher spatial consistency in both location and intensity, with relatively stable entropy variations. When comparing saliency maps across different categories, significant trend changes in model attention should emerge as DR severity increases. For instance, in mild lesions, the model may focus on fewer concentrated areas, whereas in severe or proliferative lesions, multiple lesion regions may attract attention, resulting in more complex saliency map structures and correspondingly higher entropy values.

The calculation formula of entropy is shown in Equation (1), where is the normalized pixel intensity of the significant figure.(1)$$H=-\sum_{i=1}^{n}p_{i}logp_{i}$$

Normalization of pixel intensity is typically achieved by scaling the range of pixel values to between 0 and 1. If an image’s pixel intensity falls within a specific range (e.g., 0 to 255), the normalized intensity can be calculated using Formula (2). Here, Ii represents the original intensity value of the i-th pixel in the saliency map, while min(I) and max(I) denote the minimum and maximum values of all pixels in the saliency map, respectively.(2)$$p_{i}=\frac{I_{i}-min(I_{i})}{max(I)-min(I)}$$

Trend correlation analysis aims to verify whether the focus areas of the model’s explainable methods change synchronously with the increase in lesions in fundus images. In lesion type selection, hemorrhagic spots and microaneurysms typically appear as small, round, dark-red spots in fundus images with clear boundaries and high contrast. These lesions exhibit stable morphology and distinct optical characteristics, allowing their extraction from real images through edge smoothing and color matching techniques without compromising structural consistency when embedded into other images. Compared to soft exudates (with blurred morphology) and neovascularization (with disorganized structures), adding hemorrhagic spots and microaneurysms proves more feasible. Neovascularization lesions often lack clear boundaries and are usually roughly localized using elliptical boxes or bounding circles in fundus images, making pixel-level localization and extraction challenging. Soft exudates typically feature indistinct edges, irregular shapes, and color distributions that closely match background tissues, lacking clear structural boundaries. Even after cropping from real images, maintaining naturalness becomes difficult, often resulting in “poor fusion” phenomena in target images. Although hard exudates possess higher brightness and prominent color features enabling accurate extraction, their clustered distribution in images makes unnatural fusion likely during synthesis due to brightness variations. Therefore, selecting hemorrhagic spots and microaneurysms as lesion additions in this study represents a practical experimental strategy under current conditions. In terms of analytical methodology, the trend correlation evaluation employs qualitative analysis to focus on the model’s response patterns to changes in lesion quantity. This approach aims to reveal how the model prioritizes critical regions under varying lesion counts. To achieve this, we utilized ScoreCAM’s white-box interpretability method to generate category activation maps. By applying mask processing and forward propagation to feature maps, ScoreCAM constructs discriminative saliency maps that effectively reconstruct the image regions the model focuses on during decision-making. With its strong visualization capabilities, this method proves particularly suitable for explanatory analysis tasks involving lesion quantity trends in our research.

As shown in Figure 4, when annotating fundus images, doctors use different colors to distinguish various pathological areas. Specifically, the red curve-enclosed area indicates firm exudates, the purple curve-enclosed area represents hemorrhagic points, the blue curve-enclosed area denotes microaneurysms, and the green curve-enclosed area signifies soft exudates.

The pathological lesions (bleeding points and microaneurysms) incorporated in this study were all derived from physician-annotated real ophthalmic fundus images. Based on the annotated information, we extracted typical lesion regions and applied image processing techniques including cropping, edge smoothing, and fusion to naturally overlay them onto normal fundus images, thereby generating test samples for trend correlation analysis. In the microaneurysm addition experiment, we selected representative microaneurysm areas from physician-annotated images and overlaid them onto normal fundus images, creating a sequence of images with progressively decreasing numbers of microaneurysms. As shown in the second row of Figure 5, the left-to-right images contained 14, 12, 10, 8, and 6 microaneurysm lesions, respectively. For the bleeding point addition experiment, we adopted the same image processing strategy to extract and screen typical hemorrhagic areas from authentic images, then sequentially overlayed them onto normal fundus images. As depicted in the first row of Figure 5, the left-to-right images contained 13, 11, 9, 7, and 5 hemorrhagic points, respectively.

In the fifth step, AOPC score, Recall and Dice were calculated to analyze the quality of the significant figure.

The AOPC score evaluates the fidelity of saliency maps to model predictions by progressively removing significant regions and observing changes in model outputs, thereby assessing whether the interpretation method accurately identifies key features influencing decision-making. The formula is shown in Equation (3), where represents the model’s prediction for the original input x, denotes the input image after removing the top k% of significant pixels, and K indicates the number of disturbance steps.(3)$$AOPC=\frac{1}{K}\sum_{k=1}^{K}\left(f(x)-f(x^{\left(k\right)})\right)$$

The Recall and Dice coefficients serve as key metrics for evaluating the consistency between model interpretation results and actual lesion regions. By quantitatively calculating the overlap between highlighted areas in the saliency map and lesion mask regions, these metrics provide validation of the interpretability of model predictions. In practical implementation, we first normalized the saliency maps generated by different interpretability analysis methods across models. Subsequently, using a predefined percentile threshold (0.9), we performed binary operations to retain only the top 10% high-response regions with significant values as the model’s “highlighted areas”.

The recall rate is used to measure the matching degree between the significant figure highlight area and the real mask area. The calculation formula is shown in Equation (4), where TP represents the number of pixels correctly highlighted, and FN represents the number of pixels not correctly highlighted.(4)$$Recall=\frac{TP}{TP+FN}$$

The Dice coefficient is used to measure the similarity between the significant map highlight area and the true mask. The calculation formula is shown in Equation (5), where A is the binary significant map, B is the binary label, and |A ∩ B| represents the overlapping pixels between them.(5)$$Dice=\frac{2\cdot |A\cap B|}{\left|A|+|B\right|}$$

## 4. Results

Through seven interpretable analysis methods, we generated saliency maps for each model to visualize the regions of focus in the images. Subsequently, disturbance curves were constructed based on these saliency maps to observe how model predictions change when specific regions are perturbed, allowing preliminary evaluation of the accuracy of interpretations from different methods. Figure 6 displays the saliency maps and disturbance curves obtained through testing for four classic models: VGG16, ResNet50, DenseNet161, and EfficientNet_b1. From top to bottom, they are DeepLIFT, Gradient, IntegratedGradients, SHAP, SmoothGrad, Grad-CAM++, and ScoreCAM. In each row, the first column displays the saliency map generated by the corresponding method, while the second column presents its perturbation curve. In the perturbation curve plot, the horizontal axis represents the percentage of perturbed pixels, and the vertical axis denotes the model’s confidence score for the target class. The saliency maps generated by different interpretation methods reflect variations in their assessment of input features ‘contributions to model decisions, demonstrating distinct computational strategies and focus areas. Disturbance curves help analyze the importance of saliency map-identified regions in predicting model outcomes: if the saliency region is accurate, the model’s confidence in target categories should decrease rapidly with increasing occlusion, showing a clear monotonically decreasing trend; if the saliency region lacks representativeness, the curve changes more gradually or fluctuates.

The variation trends of perturbation curves show significant differences when combined with various models and interpretability methods. As illustrated in Figure 6, both Grad-CAM++ and ScoreCAM methods exhibit pronounced fluctuations across all models. Compared to DeepLIFT, Gradient, Integrated Gradients, SHAP, and SmoothGrad, these two approaches demonstrate notable shortcomings in accurately identifying the model’s true regions of interest. Specifically, the seven perturbation curves of EfficientNet_b1 show the most dramatic fluctuations among the four models, indicating relatively poor overall interpretability. For VGG16, the perturbation curves generated by SHAP and SmoothGrad exhibit a “initial slight increase followed by rapid decline” pattern. This trend suggests that during the initial perturbation phase, the saliency maps failed to precisely cover the core regions critical for model decision-making, likely due to interference from secondary features or image noise. However, as occlusion expanded, the model’s prediction confidence rapidly declined, demonstrating that the saliency maps still effectively encompass key decision areas. Although SHAP and SmoothGrad provide relatively usable interpretability results for VGG16, their prioritization of region selection may not effectively highlight the model’s most sensitive areas. In the ResNet50 and DenseNet161 models, except for the disturbance curves generated by Grad-CAM++ and ScoreCAM interpretation methods, the saliency maps of other methods began to significantly decrease at the initial stage of disturbance, showing a stable monotonic decreasing trend. This indicates that the models can accurately locate key regions, and the interpretation results are consistent across multiple methods.

Trend correlation analysis aims to evaluate a model’s ability to perceive changes in the number of lesions within an image. Specifically, it examines whether regions of interest identified through interpretable methods show corresponding trends as the number of microaneurysm lesions gradually increases in fundus images. In this study, normal fundus images were modified by adding 14, 12, 10, 8, and 6 microaneurysm lesions, respectively, with corresponding saliency maps generated using the ScoreCAM method. Warm-colored areas in the saliency map (e.g., red and yellow) indicate regions that have a positive or critical contribution to the model’s decision-making, while cool-colored areas (e.g., blue and purple) represent regions with weak or negligible influence on the model’s judgment. As shown in Figure 7, the saliency-focused regions of the VGG16 model progressively concentrated on newly added lesion areas as the number of lesions increased. Similarly, for hemorrhagic points, this study added 13, 11, 9, 7, and 5 hemorrhagic points to another normal fundus image, generating saliency maps using the same methodology. Figure 8 demonstrates that the model’s focus areas also adjusted proportionally with the number of hemorrhagic points, validating its effectiveness in perceiving changes in lesion perception. Additionally, other models exhibited similar shifts in saliency-focused regions during trend correlation analysis, indicating that mainstream models possess some capacity to detect variations in lesion quantity.

Through preliminary analysis of disturbance curves from significant maps generated by various interpretability analysis methods, it was found that the disturbance curves of Grad-CAM++ and ScoreCAM exhibited significant fluctuations and unstable results. Therefore, in the quality assessment of significant maps, the results from these two methods were excluded to avoid interference or errors in the conclusions. This study evaluated significant map quality through four dimensions using four indicators: entropy value, AOPC score, recall rate (Recall), and Dice coefficient. The calculation formula for entropy is shown in Equation (1), which is used to measure the concentration of the model’s attention regions. A lower value indicates more focused attention and stronger model interpretability. The AOPC score calculation formula is shown in Equation (3), reflecting the average decline in classification confidence after progressively perturbing salient regions. A higher score indicates a more significant impact on model decision-making. The calculation formulas for recall and the Dice coefficient are shown in Equations (4) and (5), respectively, which are used to evaluate the consistency between salient regions and actual lesion areas, thereby reflecting the alignment between model interpretation results and ground-truth lesions.

The detailed analysis results are presented in Table 4, Table 5, Table 6, Table 7 and Table 8. These tables respectively demonstrate the average outcomes of significant figure quality analysis across five models when processing fundus images labeled as 0, 1, 2, 3, and 4. Each column in the tables presents the mean values of significant figures generated by the models using different interpretability analysis methods: DeepLIFT, Gradient, Integrated Gradients, SHAP, and SmoothGrad.

Table 4 shows the results of significant map quality analysis on the fundus images with five labels of 0 in each model. Since the mask of fundus images with labels of 0 is a full black image without real lesion areas, there is no need to analyze Recall and Dice indicators.

Overall, the VGG series models demonstrated superior performance in significant graph entropy values, with generally lower entropy levels compared to other models. Their generated significant graphs exhibited more focused attention regions. Notably, VGG16 achieved the highest AOPC score (0.914), where its significant regions significantly influenced classification decisions. The ResNet series models performed well overall in AOPC scores, particularly ResNet34 (0.876) and ResNet18 (0.853), which ranked second only to the VGG series. However, these ResNet models showed higher entropy values ranging from 14.6 to 14.9, indicating relatively scattered attention regions. DenseNet’s generated significant graphs exhibited dispersed attention information, though DenseNet169 maintained a certain advantage in AOPC. As shown in Table 4, the DenseNet series models performed averagely in significant graph entropy values, with entropy ranges between 14.63 and 14.79 showing relatively scattered attention regions. Their AOPC scores also varied significantly, exemplified by DenseNet169’s AOPC reaching 0.867 while DenseNet161 scored relatively low at 0.546. The EfficientNet series models generally exhibited higher entropy values (14.60–14.86) with scattered attention regions, showing substantial differences in AOPC scores. For instance, EfficientNet-B2 (0.611) significantly outperformed EfficientNet-B1 (0.214) and B0 (0.249). This shows that most EfficientNet models fail to form a centralized and effective significant region.

Table 5 presents the quality analysis results of saliency maps generated by various models on five labeled eye fundus images. The VGG series demonstrated overall excellence, with VGG19 outperforming in all four metrics: the lowest entropy (13.49), highest Recall (0.511), highest Dice coefficient (0.0248), and highest AOPC score (0.871). This indicates that its saliency maps not only focus on region concentration but also emphasize high overlap with actual lesion areas. VGG16 followed closely behind VGG19, performing well across all four metrics. The ResNet series maintained consistent performance on labeled images, particularly ResNet50 which excelled in Recall (0.465), Dice (0.0227), and AOPC (0.622), generating saliency maps that effectively cover true lesion regions. ResNet152 and ResNet34 showed similar performance, while ResNet18 exhibited the highest entropy (15.03), indicating relatively scattered region focus. The DenseNet series performed well in Recall and Dice metrics, exemplified by DenseNet161’s high Recall (0.439) and Dice (0.0203), demonstrating effective coverage of lesion areas. However, its generally higher entropy (approximately 14.6–14.9) suggests less focused region attention compared to the VGG series. In the AOPC scores, DenseNet121 demonstrated the best performance (0.804), indicating that its saliency regions significantly influence model predictions. In contrast, the EfficientNet series generally underperformed on labeled image 1. These models exhibited higher entropy values (14.4–14.9) with scattered saliency focus areas. They also showed relatively lower Recall and Dice coefficients. For instance, EfficientNet-B1 achieved a Recall of 0.251 and a Dice coefficient of 0.0104, while maintaining the lowest AOPC score (0.137). Consequently, the saliency regions generated by these models had minimal impact on decision-making processes.

Table 6 presents the evaluation results of significant map quality across five labeled 2-lane fundus images for various models. The VGG series demonstrated strong interpretability, with VGG19 and VGG16 outperforming others in Recall (0.4386/0.0499) and Dice (0.4289/0.0497), achieving significantly higher accuracy. Their generated maps better captured actual lesion areas. Both models maintained the lowest entropy values (VGG19:13.98 vs. VGG16:13.99), indicating highly concentrated salient regions. The AOPC scores also remained elevated, suggesting these regions significantly influenced model predictions. ResNet series models showed overall stability in labeled image performance. ResNet50 achieved optimal Dice coefficient (0.0380) and Recall (0.3337), demonstrating high consistency between maps and true lesions with excellent interpretability. ResNet152 and ResNet34 balanced multiple metrics including Recall, Dice, and AOPC, reflecting good interpretability. DenseNet models performed well in both Recall and Dice metrics. For instance, DenseNet161 achieved Recall (0.2906), Dice (0.0305), and AOPC score (0.661), indicating maps with sufficient discriminative power and coverage. However, the entropy values remain generally high (14.65–14.88), indicating that the saliency map focuses on scattered regions. While some models like DenseNet121 demonstrate better recall rates (0.3532), their Dice coefficient (0.0178) is slightly lower. Although these models achieve broader saliency map coverage, they lack sufficient localization accuracy. The EfficientNet series still shows suboptimal interpretability performance, with most models exhibiting entropy values exceeding 14.7. Additionally, both recall and Dice coefficients remain relatively low overall, particularly evident in EfficientNet-B1 where its AOPC score drops to just 0.0865—significantly below other models—suggesting minimal influence of saliency map regions on classification decisions. Although EfficientNet-B3 shows slight improvement, its overall performance remains inadequate.

Table 7 presents the significant figure quality analysis results across five retinal images labeled as category 3 (severe lesions) for various models. The VGG series continues to demonstrate exceptional interpretability, with VGG16 achieving the highest recall rate of 0.3662 and Dice score of 0.1044, indicating its ability to comprehensively cover lesion areas. VGG19 also performs remarkably well, ranking among the top with a recall rate of 0.3131 and Dice score of 0.0892. The VGG series maintains low entropy values (e.g., VGG16:14.13; VGG19:14.04), suggesting focused region selection. Notably, VGG19 BN achieves the highest AOPC score (0.7796), highlighting the critical role of significant figure regions in model decision-making. ResNet series models show relatively stable interpretability performance, particularly ResNet50 which ranks among the best mainstream architectures with a recall rate of 0.2537 and Dice score of 0.0749. ResNet152 also performs well (recall 0.2275, Dice 0.0709). Both models exhibit lower entropy values than most DenseNet and EfficientNet variants, indicating relatively concentrated region selection. In terms of AOPC scores, ResNet152 (0.647) slightly surpasses ResNet50 (0.492), suggesting greater contribution of interpretable regions to prediction accuracy. The DenseNet series models generally exhibit low Recall rates (approximately 0.15–0.18) on label-3 images, with their AOPC scores remaining consistently low. These salient regions demonstrate limited influence on model predictions. Models with higher entropy values (nearly 15) show relatively scattered feature attention. The EfficientNet series continues to underperform in interpretability metrics for label-3 images. Although Dice and Recall scores show modest improvement (e.g., EfficientNet-B2’s Dice score reaches 0.0462), their AOPC scores remain subpar (B1 at 0.1293 and B0 at 0.4344), indicating minimal contribution from these regions to decision-making. The high entropy values (mostly exceeding 14.8) suggest widespread feature dispersion. Overall, while the EfficientNet architecture demonstrates excellent computational efficiency, its interpretability remains relatively weak.

Table 8 presents the significant map quality analysis results of various models on five labeled 4-lane fundus images. The VGG series models demonstrated outstanding interpretability in labeled 4-lane fundus images, with VGG16 achieving the highest Recall (0.2766) and Dice (0.1297), significantly outperforming other models. This indicates that its generated significant maps closely align with actual lesion areas, showcasing excellent localization accuracy and coverage capabilities. VGG19 also performed well (Recall 0.2650, Dice 0.1243), maintaining the lowest entropy value (14.01) while demonstrating the most focused attention regions. VGG19 BN achieved an AOPC score of 0.6971, where the significant map’s focus area significantly influenced model predictions, indicating strong interpretative responsiveness. ResNet series models generally performed well in Dice metrics, such as ResNet152 (Dice 0.1045) and ResNet50 (Dice 0.0952), with their significant maps covering multiple lesion areas. However, their AOPC scores were relatively low (maximum 0.559 for ResNet50), suggesting limited contribution of these focus areas to model predictions. Their entropy values ranged moderately high (14.6–14.9), with relatively scattered focus areas, showing lower interpretive concentration compared to the VGG series. The DenseNet series models demonstrated strong performance in Recall and Dice metrics. For instance, DenseNet121 achieved a Dice score of 0.1014, indicating its saliency map exhibits robust lesion recognition capability. However, these models generally exhibited high entropy values (approximately 14.8–14.9) with scattered focus regions. DenseNet201 scored 0.711 in AOPC, where salient regions significantly influenced model decisions. The EfficientNet series continued to underperform on images labeled as 4. Both Recall and Dice scores remained relatively low, exemplified by EfficientNet-B3’s 0.1356 Recall and 0.0776 Dice score, along with generally lower AOPC scores.

To more intuitively present the quantitative analysis results of each model’s interpretability, we plotted line graphs based on data from Table 4, Table 5, Table 6, Table 7 and Table 8, allowing for clearer visualization of the performance differences in interpretability across different models. As shown in Figure 9, the horizontal axis indices 1 to 16 correspond to the following 16 models: DenseNet201, DenseNet169, DenseNet161, DenseNet121, EfficientNet-B3, EfficientNet-B2, EfficientNet-B1, EfficientNet-B0, ResNet152, ResNet50, ResNet34, ResNet18, VGG19-BN, VGG19, VGG16-BN, and VGG16. The labels “Label: 0” to “Label: 4” in the figure represent five categories of lesion severity: Normal, Mild Non-Proliferative Diabetic Retinopathy (Mild NPDR), Moderate NPDR, Severe NPDR, and Proliferative Diabetic Retinopathy (PDR). From the entropy comparison in Figure 9a, it can be observed that the VGG series of models perform best in terms of saliency map entropy, generally producing saliency maps with lower entropy values, which indicates more concentrated attention regions in these models. In contrast, the ResNet, DenseNet, and EfficientNet series exhibit relatively higher entropy values, suggesting more dispersed attention distributions. Figure 9b shows the performance of each model in terms of the recall metric. The VGG series achieves the highest recall, while the ResNet and DenseNet series also demonstrate relatively high recall rates. The EfficientNet series, however, shows comparatively lower performance on this metric. Based on the AOPC scores in Figure 9c, the VGG, ResNet, and DenseNet series all perform well, whereas the EfficientNet series has relatively lower AOPC scores. Figure 9d displays the performance of each model in terms of the Dice coefficient. The VGG series performs best on this metric, with the ResNet and DenseNet series also maintaining high Dice values, while the EfficientNet series shows relatively weaker performance.

## 5. Discussion and Future Research

### 5.1. Discussion

In this section of the study, we analyze and discuss the interpretability differences between four architectural models on test images. Observations of the saliency maps and perturbation curves obtained through testing for the four neural network models—VGG16, ResNet50, DenseNet161, and EfficientNet_b1—reveal that the perturbation curve of EfficientNetV2 exhibits significant fluctuations and unstable interpretation results. This may be attributed to its highly lightweight design, which employs numerous modules that compress feature dimensions, resulting in scattered saliency maps that struggle to accurately focus on critical regions. The perturbation curves of VGG16 under the SHAP and SmoothGrad methods show a “rise first, then rapid decline” trend, indicating that saliency maps effectively cover model decision areas. The simple structure and clear feature extraction of VGG16 facilitate effective interpretation methods. In contrast, the perturbation curves of ResNet50 and DenseNet161 demonstrate steady declines across most methods, with saliency maps accurately reflecting model focus areas. Their residual connection or dense connection architectures help stabilize gradient propagation, thereby enhancing interpretability. Overall, models with simple structures and clear information transmission tend to produce stable, concentrated saliency maps, while complex, compact models exhibit limitations in interpretability tasks.

The trend correlation analysis results indicate that all models exhibited a dynamic adjustment of their saliency maps in response to changes in hemorrhagic points and microaneurysm counts as lesion numbers progressively increased. Although significant structural differences exist among the models, most deep neural network models demonstrated partial lesion perception capabilities after proper training and interpretive method analysis, effectively reflecting variations in image lesion quantities. From a clinical perspective, the ability of model saliency maps to adapt to lesion quantity changes serves as a crucial indicator of interpretability. In medical-assisted diagnosis scenarios, whether models can adjust according to lesion counts not only affects the reliability of diagnostic conclusions but also directly influences physicians’ trust in model outputs.

Building on the analysis of model perturbation curves and trend correlations, this study evaluated the interpretability of four mainstream neural networks (VGG, ResNet, DenseNet, and EfficientNet) using four metrics: entropy value of saliency maps, AOPC scores, Recall, and Dice. The results demonstrate that differences in network architecture significantly impact the quality of interpretation outcomes.

The VGG series models demonstrated the best performance across all test images, with their generated saliency maps showing stable perturbation curves that accurately cover critical decision regions. This is primarily attributed to VGG’s simple architecture and clear hierarchical structure, which facilitates gradient-based and saliency map interpretation methods (such as SmoothGrad and Grad-CAM++) in identifying important areas. Among specific models, VGG16 and VGG19 exhibited the most outstanding interpretability results. In contrast, VGG16-bn and VGG19-bn slightly underperformed due to the introduction of batch normalization layers, though they still outperformed non-VGG architectures. This difference may stem from the impact of batch normalization layers on feature extraction within the model. While BN layers standardize features during training to enhance model stability and generalization, they may also weaken local sensitivity and interpretability of feature representation. Specifically, BN applies normalization across channel dimensions, which suppresses response intensity in certain feature channels. This leads to issues like weakened responses, blurred saliency maps, or scattered focus areas when interpreting important regions using gradient-based methods. Such effects are particularly evident in medical imaging tasks requiring the capture of subtle local structural variations.

The ResNet series models demonstrate overall performance, achieving high scores in AOPC and Dice metrics. Their significant overlap between saliency maps and lesion regions reflects strong localization capabilities. The residual connection architecture effectively mitigates gradient vanishing and enhances deep feature representation. However, some shallow layers (e.g., ResNet18) exhibit non-zero Recall and higher entropy values in lesion-free images, indicating potential misattention to non-critical areas. This suggests that their feature suppression capability requires further improvement.

The DenseNet series demonstrates strong performance in Recall and Dice metrics, showing potential in lesion localization. However, its significant graph entropy values tend to be relatively high, with scattered focus regions. This may be attributed to the dense connection mechanism. While this architecture enhances feature reuse efficiency and improves classification performance, the highly integrated feature channels also increase the difficulty of interpretation methods focusing on specific regions, resulting in insufficient concentration of significant graph spatial distribution.

The EfficientNet series models demonstrate the poorest interpretability across various image datasets, primarily characterized by high saliency map entropy, low Recall and Dice scores, and significant fluctuations in AOPC scores. This may be attributed to their composite scaling strategy and lightweight design. While prioritizing computational efficiency, the architecture compresses model feature representation, resulting in sparse features at intermediate layers that hinder interpretation methods from extracting stable and clear focus regions. Additionally, the limited spatial information retention capability of the fused-MBConv module further compromises the saliency map’s focalization effectiveness.

In summary, models with simple architectures and clear feature extraction demonstrate superior performance in explainability analysis, particularly in medical imaging tasks requiring high positioning accuracy. While deep or lightweight models offer certain performance advantages, they still face inherent structural limitations in explainability.

### 5.2. Future Research

This study focuses on the interpretability of diabetic retinopathy classification models, evaluating their explanatory performance on fundus image datasets and exploring the impact of model structural differences on interpretability, with the aim of enhancing the usability and trustworthiness of these models in medical practice. However, several limitations remain in this research. First, although the constructed dataset was sourced from six different medical institutions and annotated by professional ophthalmologists for both disease grading and pixel-level lesion labeling, it still exhibits certain discrepancies compared to real clinical settings. Future work should validate the stability and generalizability of the proposed interpretability evaluation scheme on larger-scale, multi-center datasets with more balanced distribution of disease severity levels. Second, the field currently lacks a unified theoretical framework for model interpretability. Although this study integrated multiple white-box and black-box interpretation methods, there is still room for optimization in the selection and combination of these methods. Future research should explore more scientific and adaptable interpretability evaluation pathways. Furthermore, this study only covered four typical architectures—VGG, ResNet, DenseNet, and EfficientNet—along with their variants, and did not include emerging models such as those based on the Transformer architecture. Subsequent work could expand the scope to more diverse model families to comprehensively reveal the influence of different structures on interpretability mechanisms. Continued exploration in these directions is expected to contribute to the development of more reliable and interpretable DR-assisted diagnostic models, promoting their safe integration and effective application in real-world clinical environments.

## 6. Conclusions

This study focuses on the interpretability of DR classification models, evaluating their explanatory effectiveness in DR images and exploring structural differences affecting model explainability. The research selected four mainstream neural network models, VGG, ResNet, DenseNet, and EfficientNet, combined with seven interpretability analysis methods including Gradient, SmoothGrad, Integrated Gradients, SHAP, DeepLIFT, Grad-CAM++, and ScoreCAM to generate saliency maps visualizing model focus regions. Subsequently, disturbance curves were constructed and trend correlation analysis was conducted, quantitatively assessing model explainability through four dimensions: entropy value of saliency maps, AOPC score, Recall, and Dice. Experimental results indicate that model architecture significantly impacts interpretability outcomes: VGG series models exhibit optimal performance with simple structures, clear feature paths, concentrated saliency map focus areas, and stable disturbance curves; ResNet models perform well in AOPC and Dice metrics but show potential misidentification risks due to shallow residual structures; DenseNet models demonstrate good Recall and Dice performance but exhibit dispersed saliency map focus and weak spatial localization capabilities; and EfficientNet models underperform in interpretability, possibly influenced by its lightweight design and sparse feature representation. Furthermore, trend correlation analysis results demonstrate that mainstream models can adapt their focus areas to evolving lesion quantities, demonstrating a certain level of lesion change perception. In conclusion, model architecture design plays a crucial role in interpretability. Models with clear structures and stable feature representation are better suited for scenarios requiring high interpretability, such as clinical diagnostic assistance. Future work could advance interpretability research for diabetic retinopathy classification models through the following three directions: First, validating the stability and generalizability of evaluation schemes in more clinically representative data environments. Second, exploring more scientific and adaptable interpretability assessment approaches. Third, expanding the scope of tested models by extending the research framework from traditional CNN architectures to emerging models such as Vision Transformer, thereby promoting the safe implementation and effective application of diabetic retinopathy classification models in real-world clinical settings.

## Figures and Tables

**Figure 1 bioengineering-12-01231-f001:**
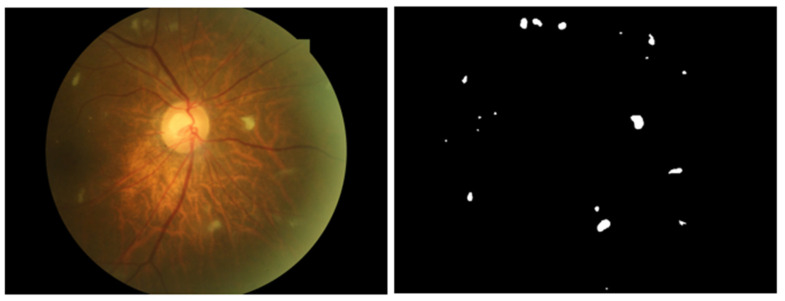
Test image and lesion mask image.

**Figure 2 bioengineering-12-01231-f002:**
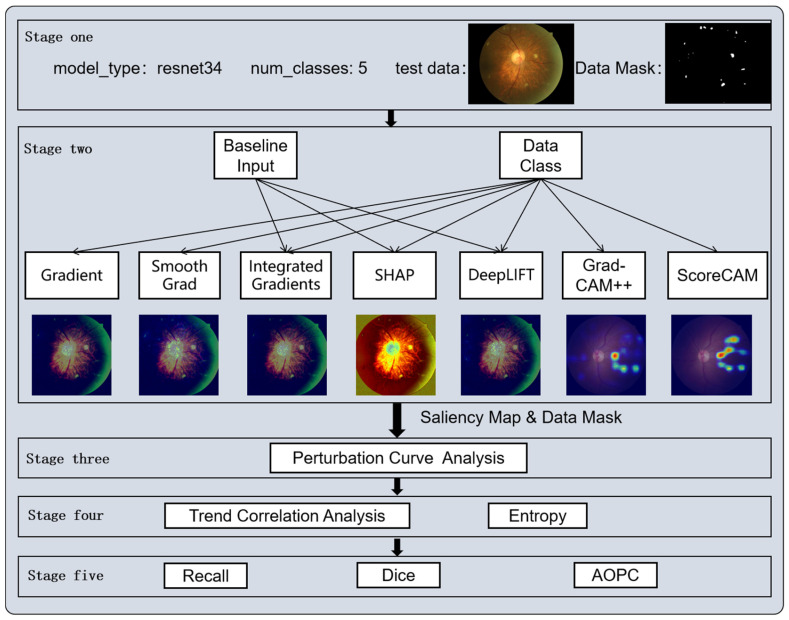
Experimental scheme.

**Figure 3 bioengineering-12-01231-f003:**
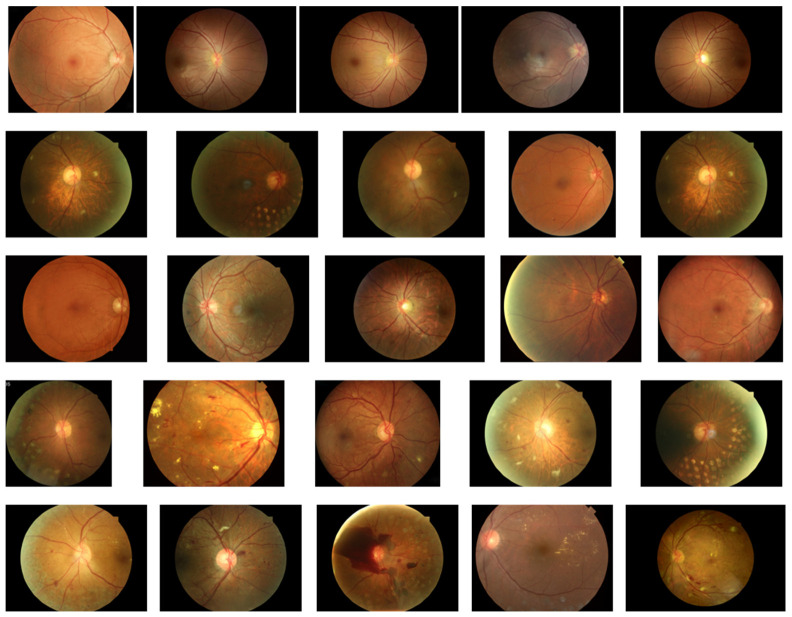
Example of images in the test set.

**Figure 4 bioengineering-12-01231-f004:**
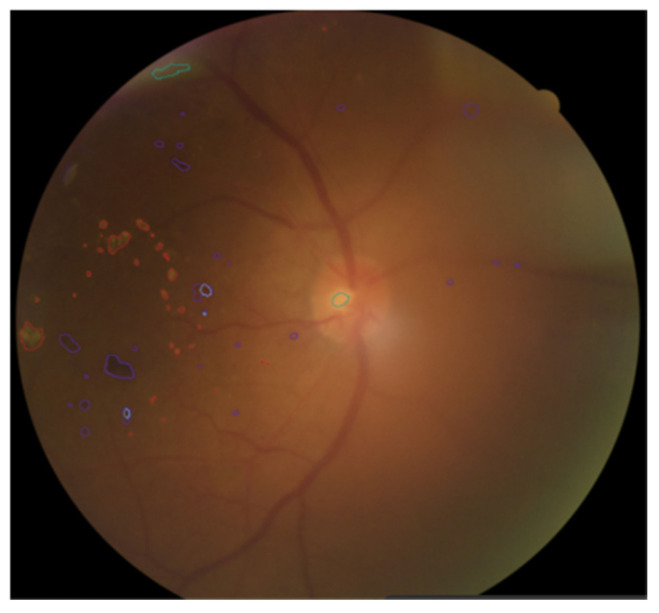
The label annotated by the clinician is a PDR fundus image.

**Figure 5 bioengineering-12-01231-f005:**
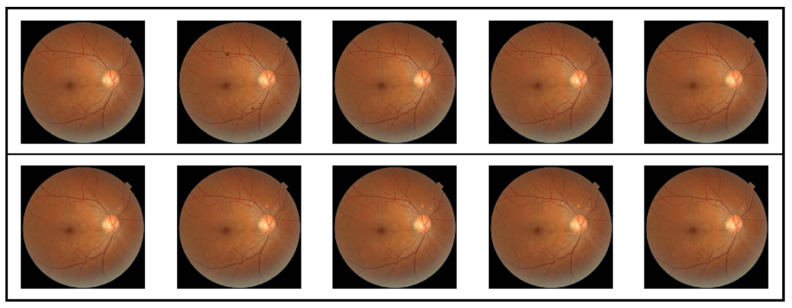
Fundus image after adding lesions.

**Figure 6 bioengineering-12-01231-f006:**
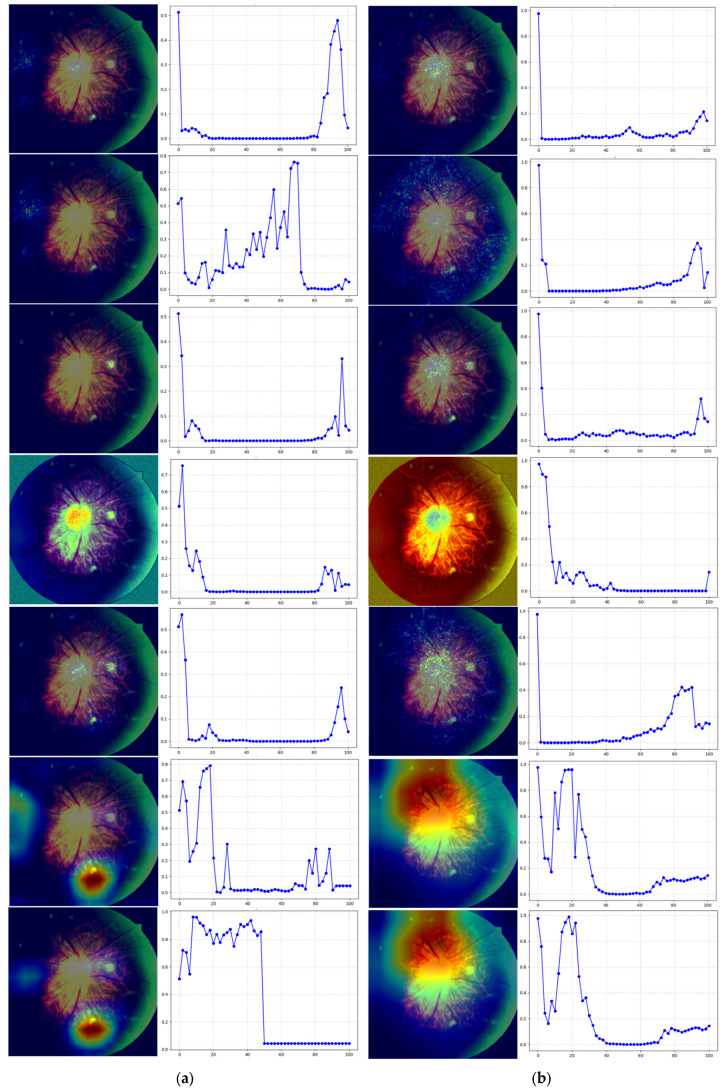
Significance plots and perturbation curves of the four models. (**a**) VGG16. (**b**) DenseNet161. (**c**) ResNet50. (**d**) EfficientNet_b1. (Perturbation Curve (Output Score vs. Perturbed Pixels): X-axis: Perturbed Piel Percentage (%); Y-axis: Model Confidence for Target Class).

**Figure 7 bioengineering-12-01231-f007:**
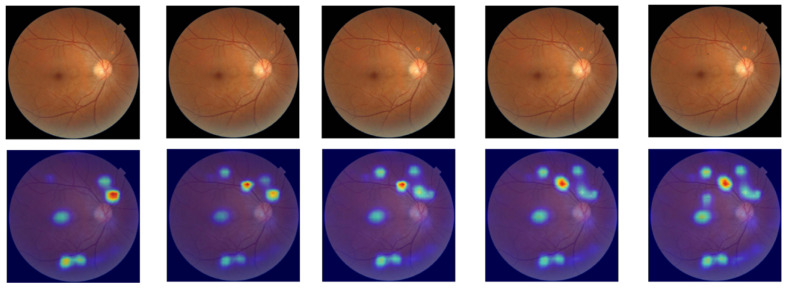
Trend correlation analysis results of VGG16 for the addition of microaneurysm lesions.

**Figure 8 bioengineering-12-01231-f008:**
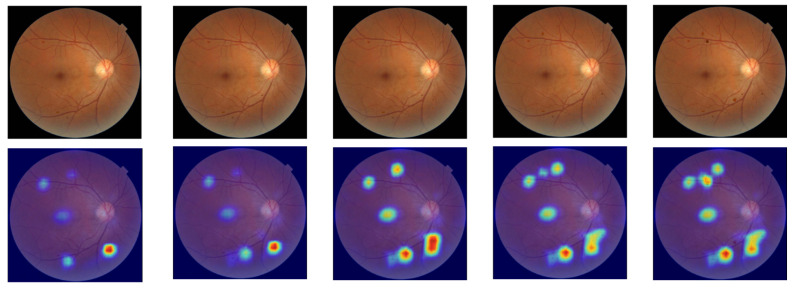
Trend correlation analysis of VGG16 with the addition of hemorrhagic lesions.

**Figure 9 bioengineering-12-01231-f009:**
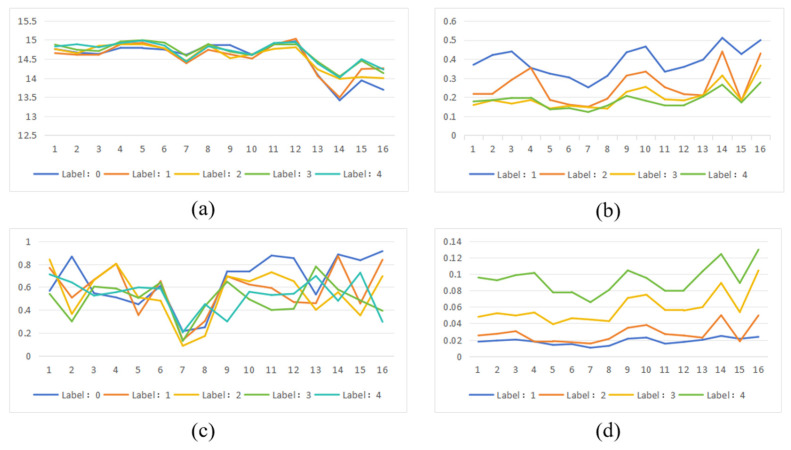
Quantitative analysis results of interpretability for each model. (**a**) Entropy. (**b**) Recall. (**c**) AOPC_Score. (**d**) Dice.

**Table 1 bioengineering-12-01231-t001:** The distribution of lesion levels in the experimental dataset.

Level of Disease	Normal	Mild NPDR	Moderate NPDR	Severe NPDR	PDR
quantity	876	167	294	183	197
proportion	51.02%	9.73%	17.13%	10.6%	11.47%

**Table 2 bioengineering-12-01231-t002:** Selected representative models from four major deep learning frameworks.

Network	Variant 1	Variant 2	Variant 3	Variant 4
VGG	VGG16	VGG16-bn	VGG19	VGG19-bn
DenseNet	DenseNet121	DenseNet161	DenseNet169	DenseNet201
ResNet	ResNet18	ResNet34	ResNet50	ResNet152
EfficientNet	EfficientNet-b0	EfficientNet-b1	EfficientNet-b2	EfficientNet-b3

**Table 3 bioengineering-12-01231-t003:** Comparison of usage methods for seven explainability analysis techniques.

Method	Require Reference Input	Specify the Image Category	Computation Complexity
Gradient	×	√	low
SmoothGrad	×	√	higher
Integrated Gradients	√	√	high
SHAP	√	√	high
DeepLIFT	√	√	middle
Grad-CAM++	×	√	low
ScoreCAM	×	√	low

**Table 4 bioengineering-12-01231-t004:** Quality analysis of significant maps of each model on five fundus images with label 0.

Model	Label	Entropy	AOPC_Score
densenet201	0	14.7599728	0.567519728
densenet169	0	14.66792297	0.866731196
densenet161	0	14.63104355	0.546402181
densenet121	0	14.79209408	0.509428256
efficientnet_b3	0	14.78351509	0.449692253
efficientnet_b2	0	14.74672649	0.611092199
efficientnet_b1	0	14.60929714	0.214488291
efficientnet_b0	0	14.86444889	0.24853842
Resnet152	0	14.86176948	0.73650479
Resnet50	0	14.61225494	0.735634422
Resnet34	0	14.90547714	0.875740894
Resnet18	0	14.96358047	0.85268137
vgg19_bn	0	14.07749027	0.535503965
vgg19	0	13.41403973	0.885490985
vgg16_bn	0	13.93613198	0.834786891
vgg16	0	13.69013462	0.914157966

**Table 5 bioengineering-12-01231-t005:** Quality analysis of significant maps of each model on five fundus images with label 1.

Model	Label	Entropy	Recall	Dice	AOPC_Score
densenet201	1	14.65259483	0.369496769	0.017742035	0.767737819
densenet169	1	14.60968147	0.420903947	0.019093763	0.507964568
densenet161	1	14.60961491	0.439191697	0.020267233	0.663370764
densenet121	1	14.88301838	0.353226086	0.017846985	0.803767908
efficientnet_b3	1	14.92519333	0.322808615	0.013787181	0.355554714
efficientnet_b2	1	14.77421955	0.302875093	0.014771375	0.65207737
efficientnet_b1	1	14.38724564	0.250757566	0.01038644	0.136793762
efficientnet_b0	1	14.73995053	0.311482077	0.012667012	0.303134489
Resnet152	1	14.62119106	0.434956704	0.021335878	0.693898344
Resnet50	1	14.51069702	0.46503258	0.022706932	0.622131135
Resnet34	1	14.87914074	0.333275217	0.015299541	0.591840984
Resnet18	1	15.02915043	0.3585871	0.017400567	0.468993459
vgg19_bn	1	14.05122368	0.396517069	0.019962856	0.458619124
vgg19	1	13.49072136	0.510730779	0.024750727	0.870569591
vgg16_bn	1	14.23583801	0.425648855	0.021253505	0.456658664
vgg16	1	14.25523224	0.499094347	0.023609441	0.838982381

**Table 6 bioengineering-12-01231-t006:** Quality analysis of significant maps of each model on five fundus images with label 2.

Model	Label	Entropy	Recall	Dice	AOPC_Score
densenet201	2	14.75965473	0.216631	0.025281411	0.840512061
densenet169	2	14.6500238	0.217763826	0.027331888	0.364721107
densenet161	2	14.84273106	0.290650031	0.030467046	0.661286431
densenet121	2	14.88301838	0.353226086	0.017846985	0.803767908
efficientnet_b3	2	14.8854507	0.185140411	0.018358445	0.509756583
efficientnet_b2	2	14.7812344	0.160046818	0.017077192	0.480533338
efficientnet_b1	2	14.44438263	0.148987328	0.015445287	0.086544762
efficientnet_b0	2	14.86780254	0.191872925	0.021186958	0.172744765
Resnet152	2	14.51949783	0.312548592	0.034568775	0.694688619
Resnet50	2	14.62611339	0.333701142	0.038000815	0.650145893
Resnet34	2	14.76475437	0.25162798	0.026923287	0.729512562
Resnet18	2	14.80539425	0.215210444	0.025185473	0.651725585
vgg19_bn	2	14.22283356	0.209132466	0.022629287	0.401238852
vgg19	2	13.97802528	0.438644754	0.049894751	0.554821887
vgg16_bn	2	14.02402592	0.180699489	0.018120721	0.352442702
vgg16	2	13.99468407	0.428861544	0.049676616	0.695412104

**Table 7 bioengineering-12-01231-t007:** Quality analysis of significant maps of each model on five fundus images with label 3.

Model	Label	Entropy	Recall	Dice	AOPC_Score
densenet201	3	14.87528217	0.158311052	0.047952933	0.540548869
densenet169	3	14.74104565	0.183002362	0.052350598	0.300058021
densenet161	3	14.70788776	0.165862474	0.049585488	0.604005585
densenet121	3	14.95691677	0.184677992	0.053189166	0.587191715
efficientnet_b3	3	14.99241577	0.140236174	0.039030576	0.505974787
efficientnet_b2	3	14.92939987	0.154164064	0.046242196	0.639920871
efficientnet_b1	3	14.5847163	0.146881589	0.044638485	0.129316668
efficientnet_b0	3	14.88966438	0.139509806	0.042713781	0.434361587
Resnet152	3	14.69075211	0.227481178	0.070860748	0.647213106
Resnet50	3	14.60255604	0.253714148	0.074863225	0.492429301
Resnet34	3	14.88159363	0.188110338	0.056237268	0.399988339
Resnet18	3	14.89816212	0.18267699	0.055593594	0.409966908
vgg19_bn	3	14.42693104	0.210533651	0.059732308	0.779598118
vgg19	3	14.04168518	0.313099812	0.089187412	0.577955677
vgg16_bn	3	14.45522385	0.182632183	0.053714265	0.483596849
vgg16	3	14.12795727	0.366163755	0.104402374	0.393069618

**Table 8 bioengineering-12-01231-t008:** Quality analysis of significant maps of each model on five fundus images with label 4.

Model	Label	Entropy	Recall	Dice	AOPC_Score
densenet201	4	14.82671745	0.177080408	0.09573731	0.711282077
densenet169	4	14.88487004	0.184332483	0.092400477	0.640911335
densenet161	4	14.80732494	0.195165187	0.098636101	0.525820279
densenet121	4	14.91785034	0.197628702	0.101408775	0.555404616
efficientnet_b3	4	14.98408825	0.135589784	0.077638534	0.597299756
efficientnet_b2	4	14.85221897	0.142078207	0.078146386	0.586335777
efficientnet_b1	4	14.43226936	0.12149864	0.065727007	0.204681933
efficientnet_b0	4	14.82967335	0.155260515	0.080731419	0.450574399
Resnet152	4	14.72008316	0.206281029	0.104466492	0.299324814
Resnet50	4	14.61024185	0.180990296	0.095243057	0.558834647
Resnet34	4	14.91924034	0.155986513	0.079688648	0.530508303
Resnet18	4	14.93994728	0.157329634	0.080341937	0.541626934
vgg19_bn	4	14.37915253	0.203113598	0.10319857	0.697136245
vgg19	4	14.0110434	0.264996232	0.12432239	0.480410927
vgg16_bn	4	14.49322968	0.171972214	0.089138554	0.725285567
vgg16	4	14.22397266	0.27659722	0.129724064	0.296522973

## Data Availability

The raw data supporting the conclusions of this article will be made available by the authors on request.

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
