# Peer review of "A Study on the Interpretability of Diabetic Retinopathy Diagnostic Models"

_bioengineering, 2025, doi:10.3390/bioengineering12111231_

Round 1

Reviewer 1 Report

Comments and Suggestions for Authors

I have read and considered the manuscript and while the research is potentially interesting there are concerns relating to the manuscript, specifically:

  1. There are acronyms in the abstract which require removal.
  2. The keywords (indexing terms) need improvement to better reflect the subject and topics addressed in this article.
  3. The introduction must be radically revised in the following dedicated sections:
    1. Introduction: setting out the background and motivation for the study, a brief overview of the proposal, the claimed contribution c/w all limitations and assumptions, and the paper structure.
    2. Related Research: the literature review and the analysis must be removed from the introduction and set out in the related research section. Moreover, the authors must provide full details of the research method for the literature review.
    3. Problem Statement: where the problem is detailed and considered c/w all potential issues, limitations, and assumptions.
  4. The M&M section should be split in to two sections:
    1. Preliminaries: where all the factors, features, and processes underlying the M&M are discussed and specified as they relate to the proposed method and all the related factors including the alternative approaches and the dataset(s).
    2. The Proposed Method (currently Materials and Methods): this section must define and describe the proposal in sufficient detail to enable the reproducibility to be evaluated.
    3. All equation numbering must be references in all related text.
  5. All references in the references section must have the DOI appended,
  6. I note the brief references to future work. However, the subject under discussion in this paper is multi-factorial with many open research questions (ORQ) drawn from both the study and the related research considered. ORQ - possibly as a sub-section in the discussion section - must be introduced and discussed c/w potential directions for future research.
  7. A principal concern relating to this manuscript lies in the focus of the study and the resulting paper. Is the principal focus of the study and the paper medical? (diagnostics) or computer science (imaging method). Such research needs a collaborative approach between computer scientists (who will have limited medical knowledge) and medical professionals (who have limited computer knowledge). There is a need for a suitable discussion relating to the collaboration implemented in reaching the conclusions drawn.

In summary, the paper presents a potentially interesting study which has the potential to be of interest to the intended audience - possibly a multi-disciplinary audience. However, there are revisions as noted above which are required to provide a logically structured document with a clear narrative flow.

Author Response

Comments 1: There are acronyms in the abstract which require removal.

Response 1: We thank the reviewer for this comment. The acronym "DR" has been removed from the abstract in the revised manuscript.

Comments 2: The keywords (indexing terms) need improvement to better reflect the subject and topics addressed in this article.

Response 2: Thank you for this constructive suggestion. We have revised the keywords to more accurately represent the core content of our study. The updated list now reads: "Diabetic Retinopathy Classification Models; Model Interpretability; Fundus Images; Interpretability Evaluation."

Comments 3: The introduction must be radically revised in the following dedicated sections:

Comments 3.1: Introduction: setting out the background and motivation for the study, a brief overview of the proposal, the claimed contribution c/w all limitations and assumptions, and the paper structure.

Response 3.1: We thank the reviewer for this guidance. The Introduction has been thoroughly revised to include dedicated sections addressing the study's background and motivation, a concise overview of the proposed approach, a clear statement of contributions , and a description of the paper's organizational structure.

Comments 3.2: Related Research: the literature review and the analysis must be removed from the introduction and set out in the related research section. Moreover, the authors must provide full details of the research method for the literature review.

Response 3.2: Thank you for this suggestion. The literature review and analysis have been moved from the Introduction to the "Related Research" section as recommended. Additionally, we have now provided a comprehensive description of the methodology used for the literature review, including search strategies, database sources, and inclusion criteria.

Comments 3.3: Problem Statement: where the problem is detailed and considered c/w all potential issues, limitations, and assumptions.

Response 3.3:Thank you for your suggestion. We have systematically elaborated the core issues addressed in this study within the main text of the paper.

Comments 4: The M&M section should be split in to two sections:

Comments 4.1: Preliminaries: where all the factors, features, and processes underlying the M&M are discussed and specified as they relate to the proposed method and all the related factors including the alternative approaches and the dataset(s).

Response 4.1: We thank the reviewer for this valuable suggestion. We have added a new paragraph before the Materials and Methods section, which outlines the datasets, models, interpretability methods, and evaluation approaches used in this study, along with an explanation of their interrelationships. This addition aims to establish a clear and comprehensive logical foundation for the detailed methodology that follows.

Comments 4.2: The Proposed Method (currently Materials and Methods): this section must define and describe the proposal in sufficient detail to enable the reproducibility to be evaluated.

Response 4.2: We thank the reviewer for this valuable suggestion. The section has been expanded to provide comprehensive methodological details to ensure the reproducibility of the study.

Comments 4.3: All equation numbering must be references in all related text.

Response 4.3: We thank the reviewer for pointing this out. All equation numbers are now properly referenced in the corresponding text throughout the manuscript.

Comments 5: All references in the references section must have the DOI appended,

Response 5: We thank the reviewer for this reminder. All references in the revised manuscript have been updated to include their corresponding DOIs.

Comments 6: I note the brief references to future work. However, the subject under discussion in this paper is multi-factorial with many open research questions (ORQ) drawn from both the study and the related research considered. ORQ - possibly as a sub-section in the discussion section - must be introduced and discussed c/w potential directions for future research.

Response 6: We have followed your suggestion by adding a "Future Research" subsection in the Discussion section. Based on several open questions identified in this study, we have elaborated on three key research directions: validation on large-scale datasets, improvement of evaluation schemes, and expansion of the range of tested models. These provide clear guidance for subsequent research. Thank you for your valuable feedback.

Comments 7: A principal concern relating to this manuscript lies in the focus of the study and the resulting paper. Is the principal focus of the study and the paper medical? (diagnostics) or computer science (imaging method). Such research needs a collaborative approach between computer scientists (who will have limited medical knowledge) and medical professionals (who have limited computer knowledge). There is a need for a suitable discussion relating to the collaboration implemented in reaching the conclusions drawn.

Response 7: We thank the reviewer for raising this important question. This study focuses on the interpretability evaluation of diabetic retinopathy (DR) classification models, with the core objective of analyzing the consistency between model decision logic and clinical cognition based on established DR medical diagnostic consensus. Adopting the perspective of AI model development, we collaborated with medical experts to construct a high-quality annotated dataset—including five-grade DR severity classifications and pixel-level lesion annotations—and designed an evaluation framework grounded in clinical diagnostic logic.At the methodological level, we applied multiple interpretability analysis techniques, including gradient-based methods (Gradient, SmoothGrad, Integrated Gradients), Shapley value-based SHAP, backpropagation-based DeepLIFT, and visual region-oriented activation mapping approaches (Grad-CAM++, ScoreCAM). Together, these methods provide multidimensional analytical perspectives ranging from input sensitivity and feature contribution distribution to attribution propagation mechanisms and visual saliency.Finally, we employed perturbation-based curve analysis and trend correlation analysis to assess the actual contribution of salient regions in the maps to model decisions and their alignment with clinical reasoning. At the same time, metrics such as entropy, recall, and the Dice coefficient were used to quantitatively evaluate the concentration of explanatory signals and their spatial alignment with ground-truth lesions.By integrating interpretability analysis with clinical diagnostic consensus, this study aims to provide a foundation for the reliable application of DR classification models in real-world medical settings and to suggest future directions for optimizing the interpretability of AI models in the field of diabetic retinopathy grading.

Reviewer 2 Report

Comments and Suggestions for Authors

Very technical study; lacks data visualization - graphs would make it easier to understand.

Author Response

Comments 1:Very technical study; lacks data visualization - graphs would make it easier to understand.

Response 1:We sincerely thank you for your valuable suggestion. We fully agree that data visualization plays a crucial role in enhancing the readability of the paper. Following your advice, we have added Figure 9 to the manuscript, which intuitively presents the quantitative analysis results of each model across four key interpretability metrics in the form of a line chart.As shown in the figure, the horizontal axis indices 1–16 correspond to 16 different models (including various variants of the DenseNet, EfficientNet, ResNet, and VGG series). The four subplots respectively represent entropy, recall, AOPC score, and Dice coefficient: Figure 9(a) shows that the VGG series performs best in saliency map entropy, with more concentrated attention distribution;Figure 9(b) indicates that the VGG series achieves the highest recall, while the ResNet and DenseNet series also demonstrate strong performance;Figure 9(c) illustrates that the VGG, ResNet, and DenseNet series all excel in AOPC score;Figure 9(d) further confirms the superior position of the VGG series in Dice coefficient.

Reviewer 3 Report

Comments and Suggestions for Authors

This is a high-quality paper that comprehensively and carefully compares multiple AI models.
However, the description of the study population is insufficient. For example, if about half of the images are from normal eyes, that would make sense if the data were collected in screening departments that only perform fundus photography for patients visiting a diabetes center. Are all six participating facilities of that kind? If so, were the subjects all diabetic patients, or did they also include individuals with no systemic disease? In which regions are these facilities located? Are they hospitals or clinics? Were any adjustments made to balance the proportions among the groups? Does the PDR group include patients after treatment whose condition had stabilized? Were the 1,717 images obtained from 1,717 eyes of 1,717 individuals? Please also describe the age and sex distribution. Have you uploaded the documentation confirming exemption from ethical review?

Author Response

Comments 1:This is a high-quality paper that comprehensively and carefully compares multiple AI models.
However, the description of the study population is insufficient. For example, if about half of the images are from normal eyes, that would make sense if the data were collected in screening departments that only perform fundus photography for patients visiting a diabetes center. Are all six participating facilities of that kind? If so, were the subjects all diabetic patients, or did they also include individuals with no systemic disease? In which regions are these facilities located? Are they hospitals or clinics? Were any adjustments made to balance the proportions among the groups? Does the PDR group include patients after treatment whose condition had stabilized? Were the 1,717 images obtained from 1,717 eyes of 1,717 individuals? Please also describe the age and sex distribution. Have you uploaded the documentation confirming exemption from ethical review?

Response 1:We thank you for your valuable comments. We have provided additional clarification regarding the data sources as follows:This study employed a retrospective random sampling method to collect fundus color photographs from ophthalmology departments at six tertiary hospitals across five different provinces in China. Geographically, the sample included three hospitals in Beijing, and one hospital each in Northeast, Southeast, and South China.Regarding the specific details of the study population, it should be noted that since this research utilizes fully anonymized retrospective data, the original dataset does not contain personal information such as patient age or gender, and therefore cannot provide relevant distribution information. Furthermore, all data used in this study contain no personally sensitive information that could identify or re-identify patients.Additionally, this study involves no commercial interests; its purpose is to investigate the interpretability of diabetic retinopathy classification models from a regulatory perspective. The research poses no physical or psychological harm to participants. It fully complies with Article 32, Paragraph 2 of China's "Ethical Review Measures for Life Sciences and Medical Research Involving Humans" (jointly issued by the National Health Commission and three other departments, implemented in 2023), which allows for exemption from ethical review. We have provided the relevant website for your and the reviewers' reference. Therefore, this study does not require ethics committee approval and is in full compliance with China's relevant research ethics regulations.

Round 2

Reviewer 1 Report

Comments and Suggestions for Authors

I have read and considered the author response and the revised manuscript and in general I am content with the revisions and the revised (highlighted) manuscript which provides enough information to enable the proposal to be evaluated.

However, there are some minor issues:

  • The manuscript format needs [albeit minor] format revisions - for example see page 15 of 19
  • I suggest that the authors reference all equation numbering is all related text.

I summary, the paper may be accepted subject to the two minor comments which may be addressed in the normal proofing process where the English grammar and manuscript format check will be completed.